# Novel Hits for N-Myristoyltransferase Inhibition Discovered by Docking-Based Screening

**DOI:** 10.3390/molecules27175478

**Published:** 2022-08-26

**Authors:** Danislav S. Spassov, Mariyana Atanasova, Irini Doytchinova

**Affiliations:** Department of Chemistry, Faculty of Pharmacy, Medical University of Sofia, 1000 Sofia, Bulgaria

**Keywords:** NMT, N-myristoyltransferase inhibitor, myristoylation, virtual screening, ZINC database, salt bridge in ligand-protein complexes

## Abstract

N-myristoyltransferase (NMT) inhibitors that were initially developed for treatment of parasitic protozoan infections, including sleeping sickness, malaria, and leismaniasis, have also shown great promise as treatment for oncological diseases. The successful transition of NMT inhibitors, which are currently at preclinical to early clinical stages, toward clinical approval and utilization may depend on the development and design of a diverse set of drug molecules with particular selectivity or pharmacological properties. In our study, we report that a common feature in the inhibitory mechanism of NMT is the formation of a salt bridge between a positively charged chemical group of the small molecule and the negatively charged C-terminus of an enzyme. Based on this observation, we designed a virtual screening protocol to identify novel ligands that mimic this mode of interaction. By screening over 1.1 million structures downloaded from the ZINC database, several hits were identified that displayed NMT inhibitory activity. The stability of the inhibitor-NMT complexes was evaluated by molecular dynamics simulations. The ligands from the stable complexes were tested in vitro and some of them appear to be promising leads for further optimization.

## 1. Introduction

The enzymes N-myristoyltransferases (NMTs) catalyze the transfer of myristic acid to the N-terminus of specific cellular proteins or peptide substrates [1,2]. Due to its hydrophobicity, the attached myristic acid is utilized for association of the modified proteins to the cellular membranes, but it also can bind to hydrophobic pockets of certain proteins, exerting control on their cellular localization or switching between on/off states [3,4,5].

NMTs use myristoyl Coenzyme A (Myr-CoA) as a cofactor to transfer myristic acid to its protein substrates [1,2]. The active site of NMT contains two adjacent pockets—one for binding to Myr-CoA and the other for binding to the substrate protein/peptide (known as the peptide binding pocket). In this configuration the N-terminal amino group of the peptide substrate comes in close proximity to the thioester bond of Myr-CoA, allowing a nucleophilic attack, which is one of the first steps in the catalytic process [1]. Coenzyme A (CoA) is released as a byproduct of this reaction [1].

NMTs are present in all eukaryotic species, including unicellular protozoan and metazoan organisms, but are absent in the prokaryotic kingdom [6]. The human genome contains two genes that encode two distinct forms of NMTs: NMT1 and NMT2 [7]. They share a very conserved catalytic domain, but their N-terminal regions are highly divergent. NMT1 and NMT2 have overlapping functions but may be regulated differently due to their unique N-terminal unstructured regions [7,8].

As myristoylation plays a key role in the life cycle of parasitic protozoan species, targeting NMTs has shown promise in the treatment of diseases such as sleeping sickness, malaria, leismaniasis, and others [9,10,11]. The first potent nanomolar inhibitors of N-myristoyltransferases were developed to target NMTs of *Trypanosoma brucei*, the causative agent of the African sleeping sickness [12]. DDD85646, the prototypical drug in this set of sulfonamide compounds, was found to be highly effective in the treatment of *Trypanosoma* infections in preclinical models [12]. IMP-1088 was developed later as a fragment-based NMT inhibitor with the aim to target *Plasmodium* spp., the protozoan species that causes malaria [10,13]. However, because the 3-dimensional structure of the catalytic domain of NMT is remarkably evolutionarily conserved, both DDD85646 and IMP-1088 were also found to be potent inhibitors of the human N-myristoyltransferases. This allows for their repurposing to treat diseases where the inhibition of human NMTs may be desirable. For example, IMP-1088 was found to block the capsid assembly and replication of cold viruses, because myristoylation of certain viral proteins by the endogenous human NMTs plays a key role in rhinovirus biology [13]. Human NMTs are also responsible for myristoylation of c-Src and c-Abl proto-oncogenes [4,14,15,16]. Myristoylation of c-Src is required for its association with cellular membranes, but also mediates its dimerization and plays a critical role in regulation of its kinase activity and function, a process that has been found to be deregulated in cancer [14,17,18]. More importantly, treatment with NMT inhibitors at concentrations that are tolerated in vivo induces apoptosis in cancer cells and has a profound effect on their overall viability [19]. In cell culture models, almost every cancer type was sensitive to NMT inhibition, including tumors from the breast, colon, pancreatic, lung, prostate, skin, and of hematopoeitic origin [19,20]. In a preclinical model, treatment with an NMT inhibitor led to the complete regression and eradication of B-cell lymphoma implanted in mice, even from a type refractory to other clinically approved treatments [20]. Therefore, NMT inhibitors, which are currently entering early stages of clinical development, are promising therapeutics that may revolutionize the treatment of some of the most dreaded diseases. The success of NMT inhibitors in clinical practice could depend on the design and development of a diverse set of drugs with specific selectivity and pharmacological or ADME (absorption, distribution, metabolism, and excretion) properties that are best suited for a particular clinical application. In this study, we describe some commonalities between the mechanisms of inhibition of the two known classes of NMT inhibitors, represented by DDD85646 and IMP-1088, and perform virtual screening to identify novel compounds with NMT inhibitory activity. The best scored compounds were tested experimentally by NMT assay and some of the hits were confirmed as promising leads for further optimization.

## 2. Results

### 2.1. A Salt Bridge between the NMT Inhibitors and the C-Terminus of NMT

At neutral pH, DDD85646 and IMP-1088 have a net charge of +1, due to protonation of a terminal nitrogen atom, part of the piperazine ring in DDD85646 and the dimethylamino group in IMP-1088 (Figure 1).

Analysis of crystallographic structures of DDD85646 and IMP-1088 complexes with human NMT1 reveals that the positively charged chemical group of the inhibitor takes part in formation of a salt bridge with the negatively charged C-terminus of the NMT protein, which is atypically located in the active site of the enzyme (Figure 2).

As DDD85646 and IMP-1088 are potent NMT inhibitors, the formation of a salt bridge with the C-terminus of NMT may play a critical role in mediating their activity. As such, we reasoned that this could be used to devise a successful virtual screening strategy that relied on identifying small molecules capable of forming similar salt bridges.

### 2.2. Virtual Screening for Identification of Novel N-Myristoyltransferase Inhibitors

#### 2.2.1. Optimization of the Docking Protocol

The crystal structure of *Homo sapiens* NMT1 in a complex with the NMT inhibitor DDD85646 (PDB 3IWE) was used to optimize the docking parameters for AutoDock Vina [21,22]. Docking the structure of DDD85646 downloaded from the ZINC15 database (with a positive charge of +1), revealed a close match with the crystallographic structure (RMSD = 1.08) and AutoDock affinity (−11.1 kcal/mol). DDD86481 is a derivative of DDD85646, and differs from it by the presence of an isobutyl group attached to its pyrazole ring [20]. With IC_50_ < 1 nM, DDD86481 is more potent than DDD85646, has 8.8-fold higher anti-tumor activity in vitro, and has been successfully used in preclinical oncological models [20]. Consistent with its higher potency, the AutoDock affinity of DDD86481 was also higher (−11.6 kcal/mol). The AutoDock affinities of DDD85646 and DDD86481 were used as a guideline to assess and compare the scores of the different ligands identified in the virtual screening. A workflow summary chart of our study is presented in Figure 3.

#### 2.2.2. Docking-Based Screening of ZINC15 Database with AutoDock Vina

The NMT inhibitors DDD85646, DDD86481, and IMP-1088 have a charge of +1 at neutral pH. Keeping this in mind, we set up a virtual screening protocol that used compounds with a charge of +1 at a pH of 7.0. Altogether, 1,114,610 structures with a predicted net charge of +1 at a pH of 7.0 were downloaded from the ZINC15 database [23] and used for docking-based screening with AutoDock Vina. The molecular weight of the selected compounds ranged from 250–500 Da and LogP values were between −1 and 5. The virtual screening identified 2560 compounds with an AutoDock affinity equal to or higher than the affinity of DDD85646 (between −11.1 and −13.4 kcal/mol) and 508 compounds with a predicted affinity equal to or higher than that of DDD86481 (between −11.6 and −13.4 kcal/mol). Of the 2560 compounds with an AutoDock affinity above or equal to DDD85646, only 55 were found to form a salt bridge with the C-terminus of NMT. In addition, it was revealed that among the hits from the virtual screening, there was a second group of compounds that formed a hydrogen bond with the C-terminus of NMT, but did not form a salt bridge because the positive charge of the molecule was located in a different moiety of the structure. A third group of compounds did not interact with the C-terminus of NMT directly. Overall, the identified ligands had diverse structures, and only rarely belonged to sets of compounds containing common structural elements.

#### 2.2.3. Prediction of ADME Properties

The large number of positive hits from the virtual screening necessitated the usage of additional filters. To narrow down the number of compounds, we selected the ones with desirable ADME properties, based on SwissADME [24]. The filters used included GI absorption—high, BBB permeability—no, violations of rules of Lipinski: 0, violations of rules of Ghose: 0–2, violations of rules of Veber: 0, violations of rules of Egan: 0, violations of rules of Muegge: 0, PAINS: 0, and Brenk: 0 (Figure 4). To eliminate ligands with potential neurotoxicity, BBB non-permeable compounds were selected at this stage only. This may be advantageous for the selection of anti-cancer therapeutics, which in most cases are BBB non-permeable. Among the 55 compounds that formed a salt bridge with the C-terminus of NMT, 28 passed the ADME filters. In addition, from the 508 compounds that showed an affinity in AutoDock equal to or higher than that of DDD86481, 79 compounds were found to have desirable ADME properties. Altogether, 107 compounds were identified with desirable ADME properties and were used for further analysis. With a few exceptions, the compounds had diverse and unique structures.

#### 2.2.4. Docking-Based Pre-Screening with GOLD

To evaluate the role of protein side chain flexibility and the presence of structural water molecules, as well as to reconfirm the docking solutions previously obtained by AutoDock Vina, we performed docking studies using GOLD software version 5.3.0 (CCDC Ltd., Cambridge, UK) [25]. All 107 molecules that passed the ADME filters were docked to human NMT1 (PDB 3IWE). The ChemPLP docking scores of the ligands ranged from 120.37 to 88.05. Fourteen compounds had docking scores higher than the score of IMP-1088, seventeen higher than the score of DDD86481, and fifty-eight higher than the score of DDD85646 (Table 1, Appendix A).

Among the top scoring compounds, the biggest proportion was of ligands that formed a salt bridge with the C-terminus of NMT. For example, 6 of the 10 top scoring ligands formed a salt bridge with the C-terminus of NMT, including ligands 1, 3, and 4 (Table 1).

The compounds identified by virtual screening represented three groups, based on their interaction with the C-terminus of NMT (Table 1, last column): compounds that form a salt bridge with C-terminus of NMT (Figure 5a,b), compounds that form a hydrogen bond with C-terminus of NMT (Figure 5c), and compounds that do not interact with the C-terminus (Figure 5d).

#### 2.2.5. Determination of Stability of Ligand-NMT Complexes by MD Simulations

35 candidate compounds and 3 control inhibitors (DDD85646, DDD86481, and IMP-1088) were chosen for determination of NMT1-ligand complex stability by molecular dynamics simulations (Figure 6, Figure 7 and Figure 8). The selected compounds were among the best hits of the virtual screening, based on GOLD and AutoDock scores, and included: (a) 17 compounds predicted to form a salt bridge with C-terminus of NMT (Figure 6); (b) 9 compounds predicted to form a hydrogen bond with C-terminus of NMT (Figure 7); and (c) 9 compounds that would not interact with the C-terminus of NMT (Figure 8).

A very good representation of the stability of the ligand–NMT complexes can be obtained by plotting the distances between selected atoms; for example, atoms participating in a salt bridge or a H-bond vs. the different frames of MD simulations. The results for the three control NMT inhibitors, DDD85646, DDD86481, and IMP-1088, indicated that the salt bridge was stable during the entire duration of the MD simulation (Figure 9).

Similarly, 15 out of the 17 ligands that formed a salt bridge with the C-terminus of NMT also formed exquisitely stable complexes with NMT1; the remaining two displayed partial stability (Figure 10).

In contrast, all nine ligands from the group that formed a hydrogen bond with the C-terminus of NMT did not form stable complexes, because the aforementioned hydrogen bond was disrupted during the MD simulation (Figure 11).

In addition, seven out of the nine compounds from the group that did not interact with C-terminus of NMT1 displayed unstable complexes (Figure 12). Altogether, the results emphasize the role of the salt bridge in the formation of stable ligand-NMT complexes.

### 2.3. Experimental Determination of Inhibitory Activity of Selected Ligands by NMT Assay

In addition to the control NMT inhibitors, DDD85646 and IMP-1088, 24 ligands were selected for experimental testing. Among them, there were 15 ligands that formed a salt bridge with C-terminus of NMT, 5 that formed a hydrogen bond with C-terminus of NMT, and 4 that didn’t interact with C-terminus. The list included ligands that displayed stable or at least partially stable complexes with NMT in the MD simulations, but a few ligands that did not form stable complexes were also included for comparison (Figure 10, Figure 11 and Figure 12). The potency of the ligands was determined by using an in vitro NMT assay (Table 2). Representative dose-response inhibitory curves are shown in Appendix A.

IC_50_ values of the control inhibitors were in the low nanomolar range (Table 2, Appendix A), in agreement with published results. For example, IC_50_ of DDD85646 has been reported as 3, 4, 13.7, or 17 nM in different studies [2,12,26,27], and in our NMT assay, it was 21.33 nM. IC_50_ of IMP-1088 has been reported to be <1 nM [13] and in our study, it was determined to be 7.61 nM (Table 2, Appendix A). However, this value is near the threshold of sensitivity of our NMT inhibition assay and may underestimate the potency of IMP-1088 (see Materials and Methods). The ligands identified by virtual screening displayed moderate activity, comparable with results from previous high-throughput screening experiments [13] (Table 2).

For example, 7 compounds showed IC_50_ in the low and middle micromolar ranges (IC_50_ < 100 µM) and 17 others had IC_50_ > 100 µM (Table 2). Five of the compounds with IC_50_ < 100 µM (ZINC19710136, ZINC67688793, ZINC19708540, ZINC19566088, and ZINC19710084) formed a salt bridge with the C-terminus of NMT; one compound (ZINC19692195) interacted with the C-terminus through a hydrogen bond; and another one (ZINC61997750) was a natural product and didn’t interact with C-terminus of NMT (Figure 13). Moreover, 7 of the 24 (29%) of the tested compounds displayed IC_50_ < 100 µM, indicating that the virtual screening had significantly amassed compounds with inhibitory activity. That the virtual screening is a technique that gathers ligands with inhibitory activity is well recognized and provided a rationale for the current study. By using this approach, we were able to identify compounds with NMT inhibitory activity, without the need to experimentally test each compound.

Structure-Activity Relationship (SAR) study revealed some common structural elements in the active compounds, such as the presence of distal benzene rings (Figure 13). Some of the most potent compounds that were identified, such as ZINC19710136 and ZINC19710084, possessed two distal benzene rings connected by an oxygen atom. Similarly, ZINC19708540 had a distal benzene ring bridged through an oxygen atom to a cyclohexane ring (Figure 13, left column). Interestingly, these substructures were not present in any of the inactive compounds. The docking poses of these ligands showed that the benzene rings inserted into a hydrophobic pocket inside the active site of NMT that formed stacking interactions with Phe188 and Phe311.

## 3. Discussion

Considering that potent NMT inhibitors are positively charged and participate in a salt bridge with the negatively charged C-terminus of NMT, we performed virtual screening of over 1.1 million ligands that had a charge of +1 at a reference pH of 7.0. Over 2500 compounds were identified that had an AutoDock affinity equal to or greater than that of DDD85646, but among them, only 55 formed a salt bridge with the C-terminus of NMT. Docking in GOLD indicated that the ligands that formed a salt bridge were among the best-scoring ligands, including the top compound from the list (Table 1). Consistent with this were also the results from the MD simulations that suggested that the ligands involved in a salt bridge with C-terminus of NMT form stable complexes (Figure 10).

The hits identified in the virtual screening were found to have more modest activity compared with the control inhibitors. This is not unexpected, considering the small probability of discovering highly active compounds directly from a virtual or, in that matter, even an experimentally based high-throughput screening (HTS). A fairer comparison would be to compare the results from the virtual screening to results from a HTS, and in this case, virtual screening outperformed previous HTS protocols tailored for NMT inhibitors. For example, the best hit discovered by HTS during development of IMP-1088 had an IC_50_ of 20 µM [13]. In comparison, the best hit from our virtual screening fairs better with an IC_50_ of 14 µM. Approximately 1/3 of the experimentally tested ligands had IC_50_ < 100 µM, indicating that the virtual screening had successfully gathered compounds with inhibitory activity; thus, providing evidence for the validity of this approach.

The ligands identified in the virtual screening may serve as leads to build more potent NMT inhibitors in the future and suggest the existence of several novel pharmacophores. For example, many hits from the virtual screening formed a salt bridge with the C-terminus of NMT through a piperidine ring, instead of a piperazine ring as in DDD85646, or a dimethylaminogroup as it is in IMP-1088 (Figure 13). Although, the piperidine and the piperazine rings are similar, they also differ in some of their properties. For instance, the piperidine is more basic (pKa 11.22) than piperazine (pKa 9.73) [28,29]; hence, it is probably better suited as a hydrogen donor in the formation of the salt bridge with the C-terminus of NMT. In addition, many of the best hits identified in the virtual screening, including ZINC19710136, ZINC67688793, ZINC19708540, ZINC19566088, ZINC19710084, and ZINC19692195 (Figure 13), had distal benzene rings that inserted into a hydrophobic pocked formed by Phe188, Phe311, Leu416, Ala418, and Val449, where they participated in sandwich π–π stacking and hydrophobic interactions. As IMP-1088 and DDD85646 do not target this hydrophobic pocket, these described distal aromatic rings could be of interest as promising novel pharmacophores.

In addition to a salt bridge with the C-terminus of NMT, IMP-1088 and DDD85646 participate in a hydrogen bond with Ser405 through their pyrazole rings (Figure 2). Thus, joining fragments with such complementary binding modes may result in the generation of more potent NMT inhibitors. In this study, we identified ZINC619997750 as a compound with micromolar activity that occupies a complementary region to the C-terminus of NMT (Figure 5 and Figure 13). Modifying its structure by introducing chemical moieties capable of forming a salt bridge with the C-terminus of N-myristoyltransferases, such as piperidine, could be a productive strategy to create novel NMT inhibitors.

In the current study, a screening and in vitro NMT assay was performed using human NMT1 protein, for several reasons. First, the development of NMT inhibitors as anti-cancer therapeutics, which is the aim of this work, restricts the targets to human NMT1 and NMT2. Second, NMT1 and NMT2 share very conserved catalytic domains, and so far, it has not been possible to selectively target only one of them, implying that the identified ligands are likely to display dual NMT1/NMT2 activity. For example, IC_50_ of DDD85646 was reported to be 17 nM against NMT1 and 22 nM against NMT2, and IC_50_ < 1 nM was reported for IMP-1088 in the assays using both proteins [2,13]. Third, it has been reported that the expression of the NMT2 gene is significantly decreased or even completely lost in certain hematological cancer types; hence, the development of a selective NMT1 inhibitor is of interest to the biotech industry [20]. Thus, in the unlikely event that some of the identified hits display selectivity against NMT1, the findings will remain relevant to the goal of development of therapeutics with anti-cancer activity.

According to SwissADME, some of the best known NMT inhibitors may have certain undesirable ADME properties. For example, DDD86481 has a violation of Lipinski’s rule of five, because it has a MW > 500 Da, a fact that may affect its tissue distribution and penetrance and reduce its efficacy as anti-cancer therapeutic, especially for treatment of solid tumors. IMP-1088 is predicted to be BBB-permeable, potentially restricting its usage as an anti-cancer therapeutic, due to possible neurotoxicity. Therefore, the design and development of novel NMT inhibitors with different pharmacological properties may play a significant role in the adoption of these novel therapeutic agents in clinical practice. The current study identifies some of the common features that mediate potency of NMT inhibitors and may facilitate future efforts in the development of these promising therapeutics.

## 4. Materials and Methods

### 4.1. Materials and Reagents

The NMT inhibitors DDD85646 (# 13839, purity ≥ 95%) and IMP-1088 (# 25366, purity ≥ 95%), as wells as ZINC214463354 (HTH-01-015, purity ≥ 95%) were obtained from Cayman Chemical (Tallinn, Estonia). ZINC19710084 (BDE23655434), ZINC19710136 (BDE23487785), ZINC19710116 (BDE23485461), ZINC19590209 (ASF18346903), ZINC19708540 (BDE26713143), ZINC19708785 (BDE26712082), ZINC72370170 (BDE23638314), ZINC19710924 (BDE25383454), ZINC19566088 (SFA21724189), ZINC19691948 (BDE25453618), ZINC19692195 (BDE25455680), ZINC72353749 (BDD26202670), ZINC257248718 (BDG33393211), ZINC19376075 (ASF19297597), and ZINC257297002 (BDH33615638) were purchased from Asinex Inc., (Amsterdam, The Netherlands). ZINC8992179 (MolPort-005-915-735), ZINC21711800 (MolPort-010-692-358), ZINC35458799 (MolPort-010-747-076) ZINC299757951 (MolPort-047-505-967), ZINC35459261 (MolPort-010-747-122), ZINC67688793 (MolPort-047-488-103), ZINC19228549 (MolPort-005-090-923), and ZINC61997750 (MolPort-002-530-610) were ordered from Molport (Riga, Latvia). The purity of the screening compounds is estimated to be at least ≥85% and in many cases to be ≥90%, according to the manufacturer’s data.

Recombinant full-length *Homo sapiens* NMT1 protein (# 80R-4067) was purchased from Fitzgerald Industries International. The substrate peptide H-Gly-Ser-Asn-Lys-Ser- Lys-Pro-Lys-NH2 used in the NMT assay corresponds to the N-terminal region of c-Src (Src peptide 2–9). The amino group at the C-terminus of the peptide indicates amidation. This sequence, as a part of the endogenous c-Src protein, is naturally myristoylated in the cells by NMTs. The peptide was custom made by GenScript. CPM (7-Diethylamino-3-(4-maleimidophenyl)-4-methylcoumarin) and myristoyl-CoA were purchased from Cayman Chemicals and the buffer components from Merck (Kenilworth, NJ, USA). The NMT assay reactions and fluorescence reading was performed in polystyrene Greiner bio 655076, 96-well black, flat bottom, medium-binding, non-sterile plates that were purchased from Merck (M4936-40EA).

### 4.2. Visualization of Protein–Ligand Interactions and Image Preparation

Protonation and protein–ligand interactions were visualized in YASARA v. 17.1.28 (IMBM, Graz, Austria) [30]. Images were prepared in PyMOL 1.6.0.0 (Schrödinger, New York, NY, USA).

### 4.3. Virtual Screening of ZINC Database with AutoDock Vina

The virtual screening was performed using the structures of 1,114,610 distinct commercially available compounds that have a charge of +1 at a reference pH of 7.0. The structures were downloaded from the ZINC15 database [23] in pdbqt format in July 2020. The filters used included: reactivity-standard; purchasability—in stock; pH-Reference, pH 7.0; charge: +1; and MW and LogP-drug-like (compounds with MW 250–500 Da and LogP between −1 and 5). The virtual screening was performed by using AutoDock Vina [21,22,31]. The ligands were docked into the X-ray structure of *Homo sapiens* NMT1 in complex with DDD85646 and Myr-CoA (PDB 3IWE). Prior to docking, DDD85646 was extracted in silico and the binding site was defined by a grid box with these coordinates: center x = −11.077, center y = 21.635, center z = −5.355, size x = 28, size y = 26, and size z = 26. Considering a published report about cooperativity in binding between Myr-CoA and NMT inhibitors [12], Myr-CoA was kept in the structure and was present during the docking. The parameters were set to energy range = 3 and exhaustiveness = 20. The docking was performed at FUJITSU servers with GPU NVIDIA Tesla V100 at the Faculty of Pharmacy, Medical University of Sofia, on a Linux operating system.

### 4.4. Determination of ADME Properties

The ADME properties were determined by SwissADME, hosted at the Swiss Institute of Bioinformatics http://www.swissadme.ch (accessed on 15 October 2020) [24], and using smiles of the ligand structures downloaded from the ZINC15 database.

### 4.5. Molecular Docking by GOLD

Docking was performed by using GOLDsuite version 5.3.0 (CCDC Ltd., Cambridge, UK) [25]. The structures of the ligands were downloaded directly from the ZINC15 database in mol2 format, with protonation state corresponding to the reference pH of 7.0. The crystal structure of human NMT1 in complex with DDD85646 and Myr-CoA (PDB 3IWE) was used for docking after removal of the bound inhibitor. The radius of the binding site was set to 6 Å. The protocol was optimized in terms of scoring function, rigidity/flexibility of amino acid side chains, presence or absence of a water molecule within the binding site, and number of genetic algorithms (GA) runs. Re-docking the structure of DDD85646 using the optimized protocol revealed a close match with the crystallographic structure (RMSD = 1.21). The docking runs in the present study were performed by using protein receptors with flexible binding sites, flexible ligands, and scoring function ChemPLP. Ten amino acid residues from the binding site, which were in close proximity to the bound ligand (Phe188, Phe311, Tyr296, His298, Y180, Phe190, Asn246, Thr282, Ser405, and Gln496), were set as flexible. Three structural water molecules (HOH2, HOH760, and HOH970) were also present and were set to toggle and spin. The docking was performed in 100 GA runs for each ligand, but only the top 10 scored solutions were saved for analysis.

### 4.6. Stability of the NMT1-Ligand Complexes by Molecular Dynamics Simulations

The best-scored docking solutions were chosen as starting frames for molecular dynamics (MD) simulations, as previously described [32]. Briefly, the ligand structures were parametrized using GAFF2.11 force field [33] and AM1-BCC charges [34] and the complex was solvated in saline in a truncated octahedral box, with energy minimized, heated to 310 °C at constant volume for 1 ns, density equilibrated at 1 bar for 1 ns, equilibrated keeping constant T and p for 1 ns, using the Langevin thermostat [35] and Berendsen barostat [36], and simulated for 1000 ns by AMBER v. 18 (UCSF, CA, USA) [37,38]. The systems were simulated with the ff14SB force field [39] under periodic boundary conditions. Frames were saved every 1 ns to generate 1000 frames. This data was analyzed in VMD (Visual Molecular Dynamics, University of Illinois, Urbana-Champaign, IL, USA) [40]. Representations of the results were obtained by plotting the distances between selected atoms; for example, atoms participating in a salt bridge or an H-bond, vs. the different frames of simulations. The results were saved as *.dat files, opened in Notepad, and then used to generate the graphs in GraphPad Prism.

### 4.7. NMT Inhibition Assay

IC_50_ values were determined by using a fluorogenic NMT assay, adopted from Goncalves et al. [27]. The assay detects Coenzyme A (CoA) that is released as a byproduct of the enzymatic reaction. Briefly, 18.9 nM of the *Homo sapiens* NMT1 enzyme was incubated in the presence of 3-fold dilutions of the various ligands, 4 µM myristoyl-CoA, 4 µM substrate peptide, and 2.7% DMSO. The reactions were performed in a reaction buffer of 20 mM potassium phosphate, pH 8.0, 0.5 mM EDTA, and 0.1% (*v/v*) Triton^®^ X-100, in 44 µL volume in a 96-well black plate. To eliminate background fluorescence for each concentration of the ligands, reactions were also performed in the absence of the NMT1 enzyme. After 30 min of incubation at 25 °C, 20 µL of CPM in reaction buffer was added to the final concentration of 5.5 µM. The reaction was stopped after 5 min by adding 36 µL of 0.1 M sodium acetate buffer (pH 4.75). Fluorescence was measured at an excitation wavelength of 380 nm and emission wavelength of 470 nm with CLARIOstar fluorescent reader (BMG LABTECH GmbH, Ortenberg, Germany). To eliminate the background signal, the fluorescent values of samples obtained in the absence of NMT1 were subtracted from the fluorescent signal of samples performed in the presence of NMT1. The subtracted values were plotted in GraphPad Prism using log(inhibitor) vs. normalized response nonlinear fit to generate inhibitory curves and determine IC_50_ values. The highest final concentration of the ligands used in the assay was 100 µM, and if this concentration was not sufficient to reduce the specific signal by 50% or more, IC_50_ > 100 µM value was assigned. In theory, the experimentally determined IC_50_ cannot be lower than half the enzyme concentration used in the assay (9.45 nM for the current assay). Therefore, IC_50_ of inhibitors with potencies that are near or below the threshold of the assay cannot be determined with accuracy. The assay was performed with full-length NMT1 protein, instead of the purified catalytic domain of NMT, as done in all previous studies. Although this may have influenced some of the kinetic parameters of the enzyme reaction, it was found that the IC_50_ of the NMT inhibitors were not substantially different from previously published results.

## 5. Conclusions

Using a fragment-based approach and seeds containing positively charged chemical groups capable of interacting with the negatively charged C-terminus of N-myristoyltransferases could be critically important for further attempts to develop novel NMT inhibitors. The ligands identified in the virtual screening could serve as such lead molecules for further optimization and development.

## Figures and Tables

**Figure 1 molecules-27-05478-f001:**
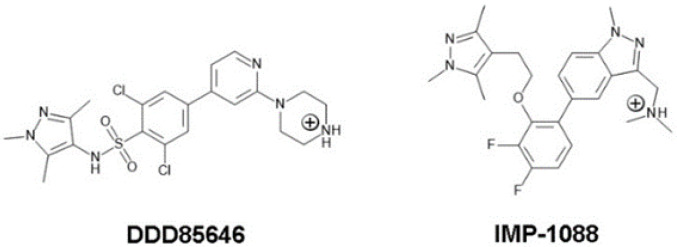
2D structures and protonation states of the NMT inhibitors, DDD85646 and IMP-1088, at a pH of 7.0. Both molecules have a net charge of +1 due to protonation of the piperazine ring of DDD85646 and the dimethylamino group of IMP-1088.

**Figure 2 molecules-27-05478-f002:**
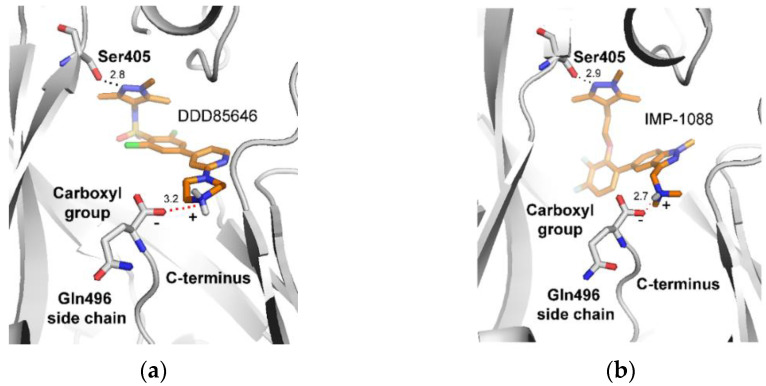
Formation of a salt bridge between the positively charged moiety of the NMT inhibitors and the C-terminus of the NMT1 protein based on: (**a**) the crystal structure of *Homo sapiens* NMT1 in a complex with DDD85646 (PDB 3IWE) [12]; (**b**) the crystal structure of *Homo sapiens* NMT1 in a complex with IMP-1088 (PDB 5MU6) [13]. The carbon atoms of the inhibitors are shown in orange. The salt bridge is depicted by red dots. The corresponding charges of the inhibitors and of the carboxyl group at the C-terminus are also indicated. Distances shown are in angstroms (Å).

**Figure 3 molecules-27-05478-f003:**
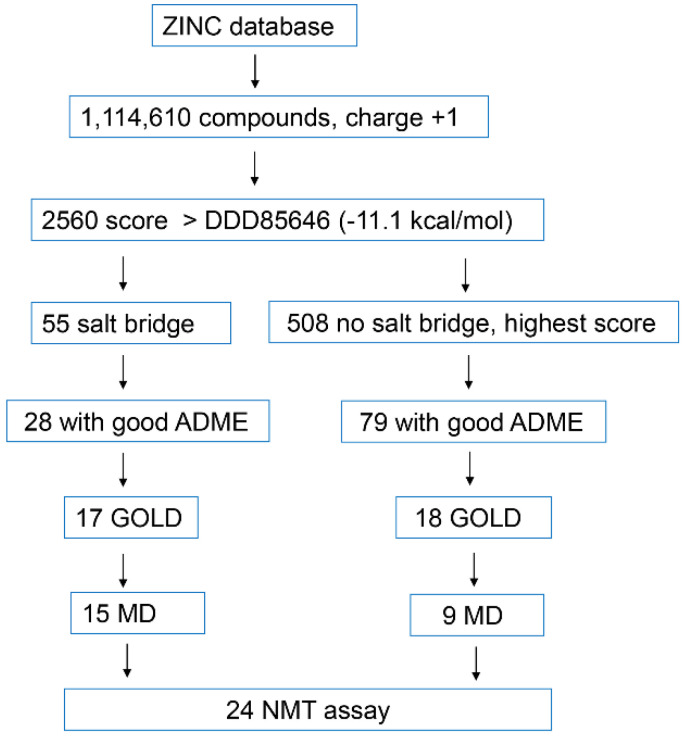
A workflow chart of the virtual screening. Over 1.1 million structures with a charge of +1 were downloaded from the ZINC database. By using AutoDock Vina, 2560 compounds were identified that had a higher AutoDock affinity score than the score of the potent NMT inhibitor, DDD85646 (−11.1 kcal/mol). Of them, 55 were found to form a salt bridge with the C-terminus of NMT. In parallel, 508 of the highest scoring ligands that did not participate in a salt bridge with the C-terminus of NMT were also analyzed. Filtering the compounds based on their ADME properties narrowed down the number of compounds to 28 and 79, respectively. These compounds were used for docking using GOLD software, and the best scoring 35 were selected (17 that form a salt bridge and 18 that do not). Molecular Dynamics (MD) was performed on the NMT complexes of these 35 ligands, and finally, 24 compounds were selected for experimental testing in an in vitro fluorescent NMT assay. The group that did not form a salt bridge with the C-terminus of NMT, comprised two types of compounds: some that formed a hydrogen bond with the C-terminus of NMT and some that did not interact with the C-terminus at all.

**Figure 4 molecules-27-05478-f004:**
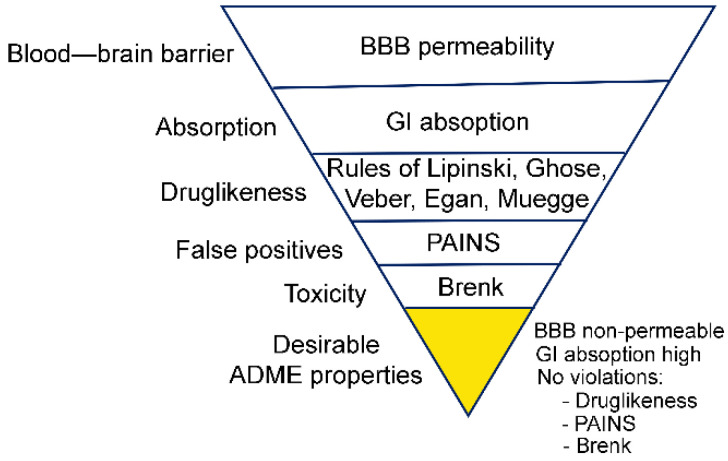
Ligand selection, based on ADME filtering.

**Figure 5 molecules-27-05478-f005:**
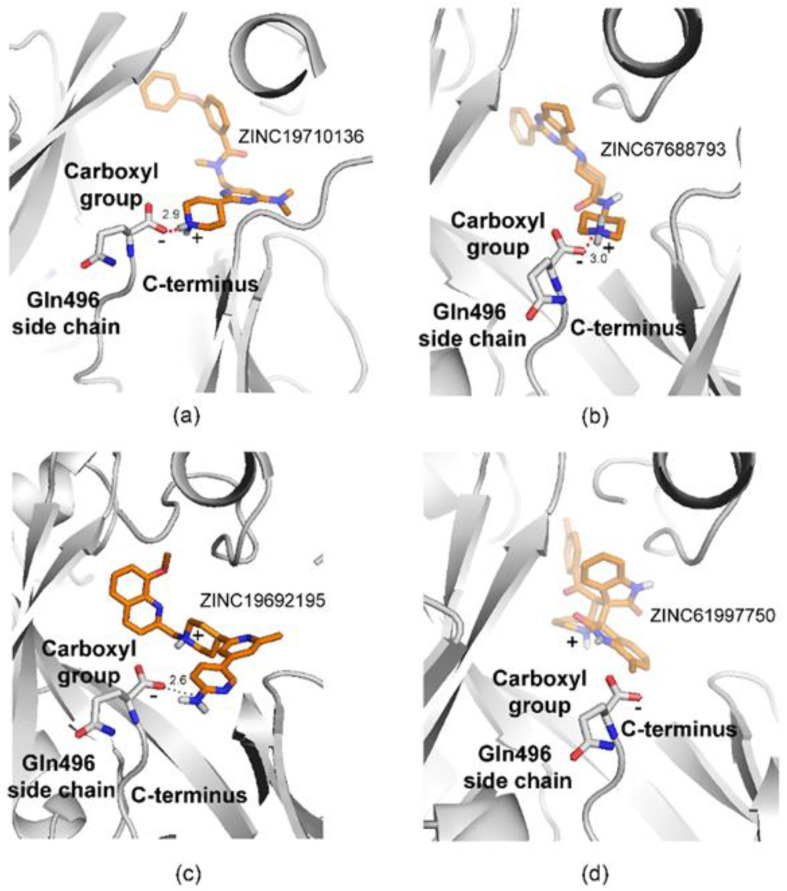
Examples of docking poses of ligands identified by virtual screening. The distances shown are in angstroms (Å). (**a**) ZINC19710136 forms a salt bridge with C-terminus of NMT1. (**b**) ZINC67688793 forms a salt bridge with C-terminus of NMT1. (**c**) ZINC1969215 participates in a hydrogen bond with the C-terminus of NMT1. The charge of the small molecule is located in a central piperidine ring and is not involved in the formation of a salt bridge (**d**) ZINC61997750 occupies a complementary region and does not interact with the C-terminus of NMT1. The docking was performed using GOLD with flexible ligand and flexible amino acid side chains settings. The images were produced in PYMOL.

**Figure 6 molecules-27-05478-f006:**
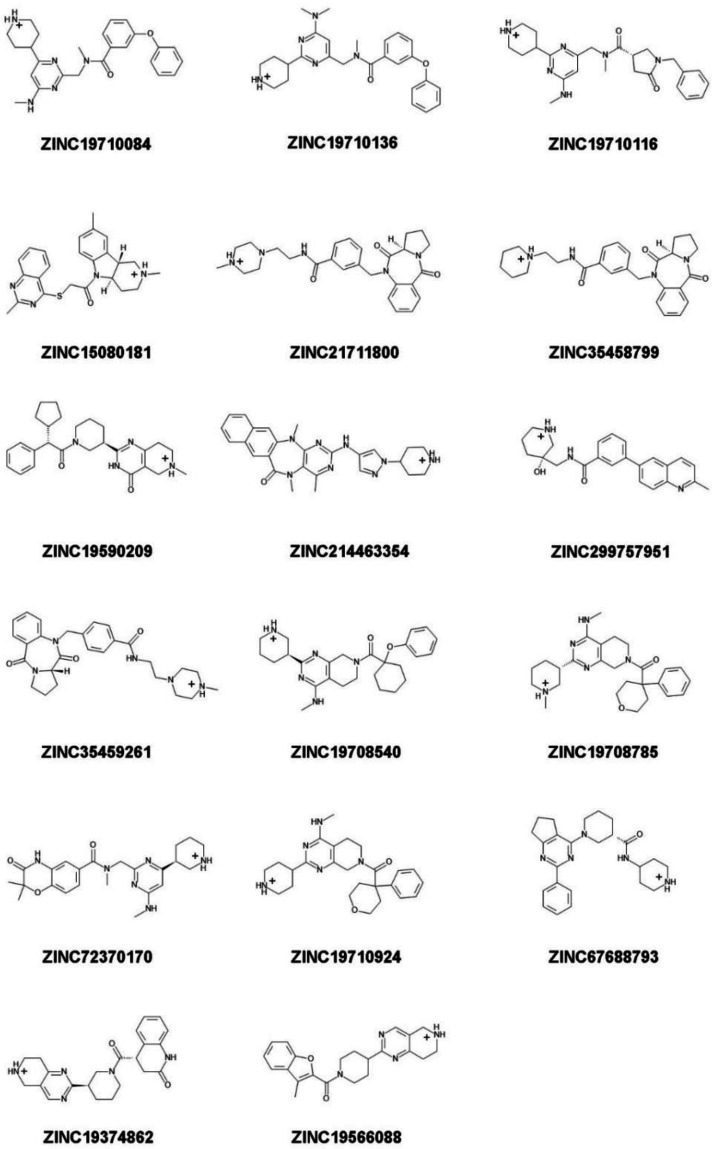
Structure, protonation, and charge at pH 7.4 of ligands identified in the virtual screening with predicted salt bridge with the C-terminus of NMT. The salt bridge is formed between the positively charged moiety of the small molecule and the negatively charged C-terminal carboxyl group of NMT protein. The positive charge of the ligand is located at a distal piperidine or piperazine ring.

**Figure 7 molecules-27-05478-f007:**
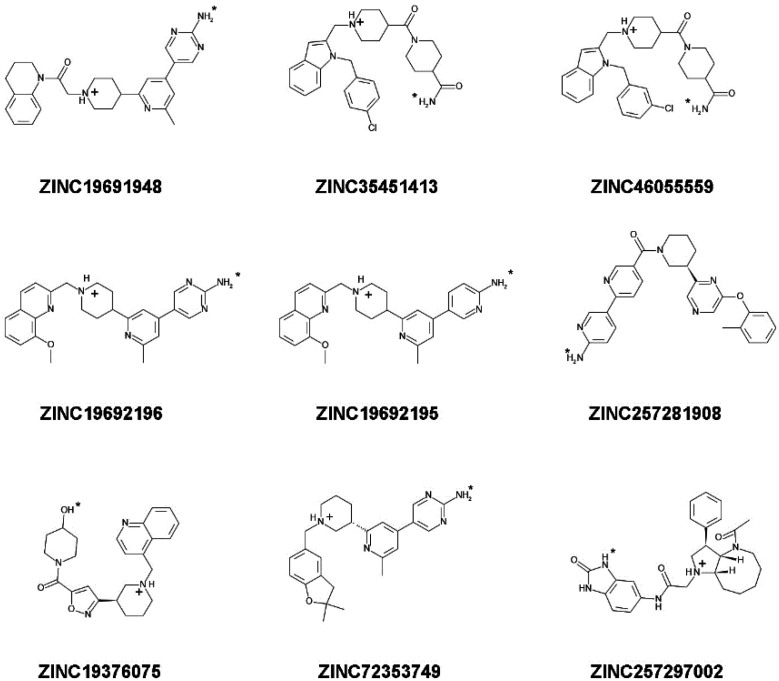
Structure, charge, and protonation at pH 7.4 of candidate ligands that form a hydrogen bond with C-terminus of NMT1. The hydrogen atom involved in the H-bond is highlighted by a small asterisk. This hydrogen atom can be a part of an amino group, hydroxyl group, or indole ring in the different ligands. The positive charge of the molecule is in a centrally located ring substructure and does not engage in formation of a salt bridge (may participate in cation–pi interactions in some cases).

**Figure 8 molecules-27-05478-f008:**
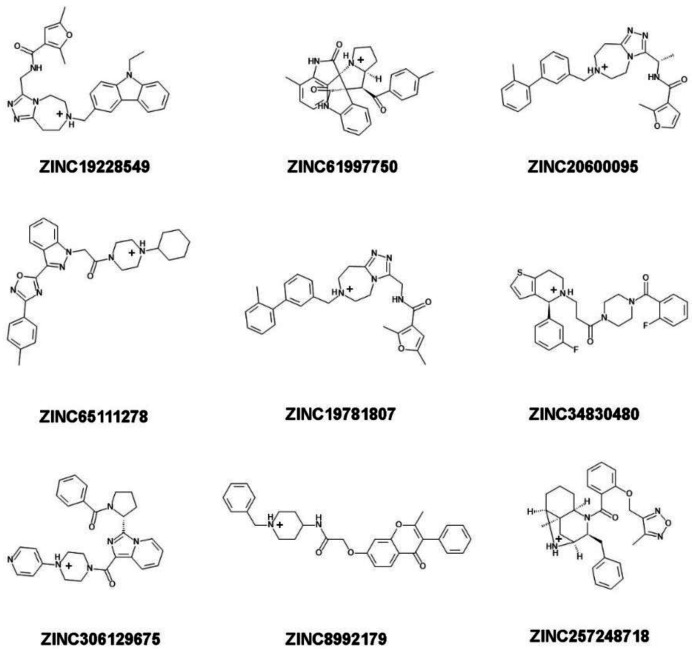
Structure, protonation, and charge at pH 7.4 of candidate ligands that do not interact with the C-terminus of NMT1. These ligands do not engage the C-terminus of NMT in a salt bridge or in a hydrogen bond.

**Figure 9 molecules-27-05478-f009:**
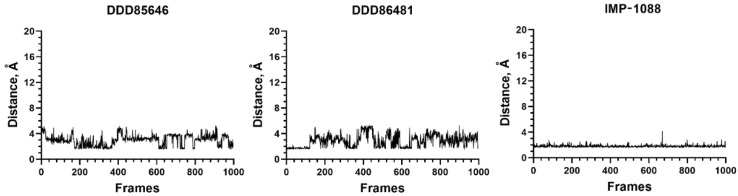
NMT inhibitors DDD85646, DDD86481, and IMP-1088 form stable complexes in MD simulations. Here the length of the salt bridge is plotted against the different frames of simulation. Note that the distances do not exceed 4 Å, indicating that the salt bridge is preserved and stable during the simulations.

**Figure 10 molecules-27-05478-f010:**
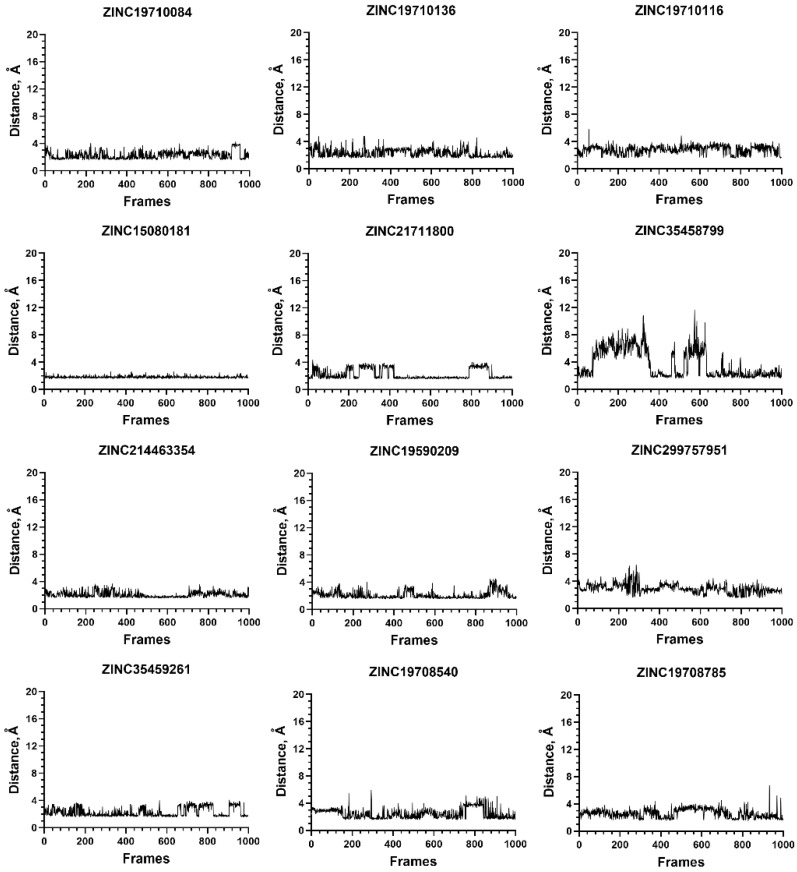
Stability of the ligand–protein complexes for compounds that form a salt bridge with C-terminus of NMT1. The graphs depict the length of the salt bridge at the different frames of MD simulations. The salt bridge is considered stable if its distance does not exceed 4 Å. The complex of ZINC353458799 (second row, right) displays partial stability; initially, the salt bridge is disrupted, but later, it is re-formed (after frame 650). The complex of ZINC67688793 (fifth row, right) is at least partly unstable, because the salt bridge becomes disrupted in frames 780–960. Altogether, the results emphasize the role of the salt bridge in formation of stable ligand–protein complexes.

**Figure 11 molecules-27-05478-f011:**
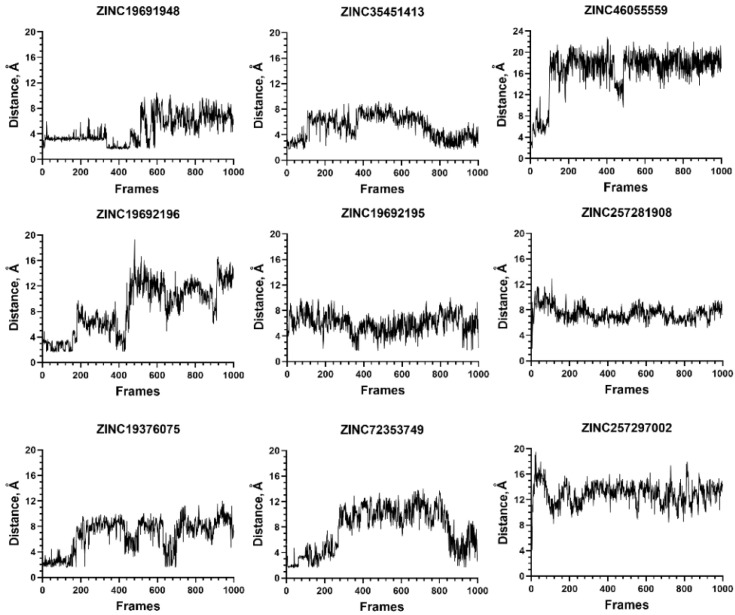
Compounds that interact through a hydrogen bond with C-terminus of NMT1 do not form stable complexes in MD simulations. The distance between atoms participating in hydrogen bonds between the ligand and C-end of NMT1 are depicted. During the MD simulation, the distances increased above 4 Å for all ligands, indicating dissociation of the hydrogen bond.

**Figure 12 molecules-27-05478-f012:**
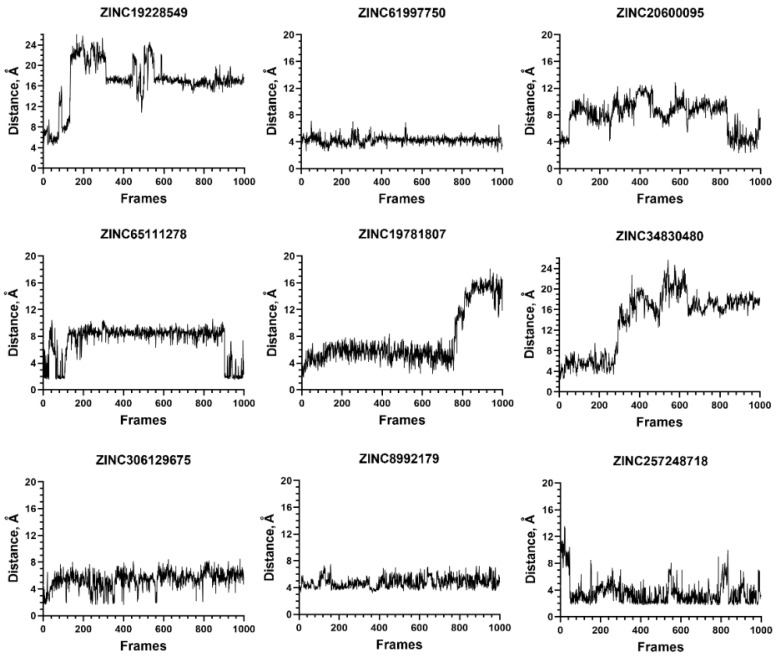
Stability of the ligand–protein complexes for compounds that do not interact with the C-terminus of NMT1. The values on the *Y*-axis show the distance between selected atoms during the MD simulation. For ZINC19228549, these were the atoms participating in a hydrogen bond with Gly284; for ZINC61997750, ZINC65111278, and ZINC257248718—the atoms involved in a hydrogen bond with Tyr420; for ZINC20600095—the atoms participating in a hydrogen bond between the ligand and coenzyme A; for ZINC19781807—the distance between the ligand and Asn473; and for ZINC34830480 and ZINC306129675—the distance to Gln496 (non-polar contact). Distance > 4 Å indicates instability.

**Figure 13 molecules-27-05478-f013:**
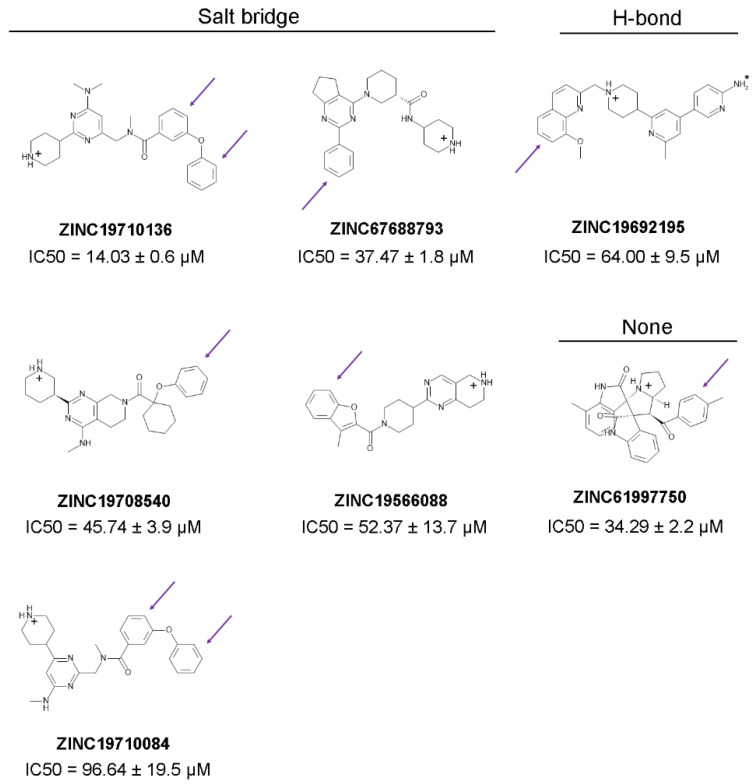
Structures of compounds identified in the virtual screening with IC_50_ < 100 µM. The predicted positive charge of the molecules at pH 7.0 is indicated by a plus sign. Five ligands form a salt bridge with C-terminus of NMT though a positively charged piperidine ring. The amino-group of ZINC19692195 (indicated by a small asterisk), forms a hydrogen bond with the C-terminus of NMT1. ZINC619997750 is a natural product that does not interact with the C-terminus of NMT. Distal benzene rings are indicated by arrows.

**Table 1 molecules-27-05478-t001:** The top scoring ligands of the virtual screening in GOLD. The ligands with ChemPLP score above the score of DDD86481 are shown. The AutoDock affinity of all ligands is equal to or higher than the one of DDD85646 (−11.1 kcal/mol). The control NMT inhibitors are shaded in grey. Complete list of ligands and their docking scores is provided in Appendix A.

	ZINC ID	ChemPLP GOLD	Affinity AutoDock kcal/mol	Interaction with C-Terminus of NMT
1	ZINC19710084	120.37	−11.4	Salt bridge
2	ZINC19691948	120.17	−12.0	H-bond
3	ZINC19710136	119.73	−11.6	Salt bridge
4	ZINC19710116	117.55	−11.3	Salt bridge
5	ZINC19228549	117.52	−11.9	None
6	ZINC15080181	116.02	−11.1	Salt bridge
7	ZINC19710117	115.26	−11.3	Salt bridge
8	ZINC35451413	114.82	−11.8	H-bond
9	ZINC21711800	113.94	−11.3	Salt bridge
10	ZINC61997750	113.09	−11.9	None
11	ZINC46055559	112.95	−11.9	H-bond
12	ZINC35458799	112.31	−11.3	Salt bridge
13	ZINC19692196	111.64	−11.8	H-bond
14	ZINC20600095	111.56	−11.9	None
ctr	IMP-1088	111.36	−10.8	Salt bridge
15	ZINC19692195	110.28	−12.0	H-bond
16	ZINC65111278	110.24	−11.6	None
17	ZINC19590209	110.20	−11.9	Salt bridge
ctr	DDD86481	110.20	−11.6	Salt bridge

**Table 2 molecules-27-05478-t002:** IC_50_ and docking scores of ligands identified in the virtual screening. The control NMT inhibitors are shaded in grey.

	ZINC ID	IC_50_, µM	ChemPLP GOLD	Affinity AutoDock kcal/mol	Interaction with C-Terminus of NMT
1	ZINC19710136	14.03 ± 0.6	119.73	−11.6	Salt bridge
2	ZINC61997750	34.29 ± 2.2	113.09	−11.9	None
3	ZINC67688793	37.47 ± 1.8	98.10	−11.4	Salt bridge
4	ZINC19708540	45.74 ± 3.9	105.16	−11.2	Salt bridge
5	ZINC19566088	52.37 ± 13.7	94.50	−11.1	Salt bridge
6	ZINC19692195	64.00 ± 9.5	110.28	−12.0	H-bond
7	ZINC19710084	96.64 ± 19.5	120.37	−11.4	Salt bridge
8	ZINC19708784	>100	101.21	−11.6	Salt bridge
9	ZINC299757951	>100	106.95	−11.5	Salt bridge
10	ZINC21711800	>100	113.94	−11.3	Salt bridge
11	ZINC19228549	>100	117.52	−11.9	None
12	ZINC35458799	>100	112.31	−11.3	Salt bridge
13	ZINC35459261	>100	105.64	−11.7	Salt bridge
14	ZINC8992179	>100	108.00	−11.6	None
15	ZINC72353749	>100	106.19	−11.7	H-bond
16	ZINC72370170	>100	102.19	−11.5	Salt bridge
17	ZINC19710924	>100	102.12	−11.4	Salt bridge
18	ZINC19691948	>100	120.17	−12.0	H-bond
19	ZINC257248718	>100	107.29	−11.9	None
20	ZINC19376075	>100	107.51	−12.2	H-bond
21	ZINC257297002	>100	105.31	−12.0	H-bond
22	ZINC19710116	>100	117.55	−11.3	Salt bridge
23	ZINC19590209	>100	110.20	−11.9	Salt bridge
24	ZINC214463354	>100	110.07	−12.0	Salt bridge
25	DDD85646	0.02133 ± 0.00008	103.59	−11.1	Salt bridge
26	IMP-1088	<0.007611 ± 0.0006	111.36	−10.8	Salt bridge

## Data Availability

Not applicable.

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
