# Peer review of "Novel Hits for N-Myristoyltransferase Inhibition Discovered by Docking-Based Screening"

_molecules, 2022, doi:10.3390/molecules27175478_

Round 1

Reviewer 1 Report

Spassov et al. submitted the manuscript, "Novel Hits for N-myristoyl transferase Inhibition Discovered by Docking-based Screening, " about designing and evaluating the NMT inhibitors. These NMT inhibitors are sighted in the interest of treating protozoa-based diseases. Authors conceptualize the salt-bridge formation between a positively charged group from an inhibitor and a negatively charged enzyme. Their observation led to exploring evaluate of new inhibitors through virtual screening. Screening of 1.1 million structures from the ZINC-database produced hits whose complex stability was evaluated by molecular dynamics.

The strength of the paper lies in the rational use of computational tools (docking and molecular dynamics) and further correlating these results with in-vitro data.

1.       The compounds were purchased from Molport (Riga, Latvia); however, the purity of these compounds should be mentioned in the manuscript. Please add a few lines regarding the purity of compounds.

2.       Please incorporate the compound structures from the supplementary information (Figure S1, S2, and S3) into the main body of the manuscript as they would improve the paper's readability.

3.       However, it seems that affinity Autodock (kcal/mol) doesn't have any proportional effect on the IC50's. Therefore, the author needs to provide a statement in the text which rationalize the current study.

4.       Structural activity relationship (SAR): There were 26 compounds, and only 7 showed activities. Were there any features/structural similarities among the active compounds compared to inactive compounds observed? It would add more value to the current study if a small paragraph stating the SAR of these compounds is added to the paper.

5.       MD simulation graphs or figures should be added to the main manuscript (at least for the most potent compound), to engage readers in the main manuscript.

Authors have written a manuscript systematically, covering all aspects of a medicinal chemistry paper, which comes into the interest of the current journal.

Reviewer 2 Report

This research paper reported the discovery of new hit compounds against N-myristoyltransferases via molecular docking and molecular dynamics (MD) simulations. The topic is interesting to the research community, and this study is of importance for the development of N-myristoyltransferases inhibitors. The manuscript is well-organized, while the necessary experiments are needed to support the results and conclusions in this manuscript.

Major points:

1. One recent report (Kallemeijn et al., Cell Chem Biol, 2019, 26, 892) showed the invalidation of 3 commonly used inhibitors of human N-myristoyltransferase via the orthogonal assays. To remove the bias of the fluorescence-based assay in the current study, the authors are required to perform the secondary binding assay (such as ITC, SPR assays or the others) with different mechanism to validate the binding and enzymatic results of the potent inhibitors in this study.

2. The off-target toxicity is a big problem to the development of N-myristoyltransferases inhibitors. The basic cellular toxicity of the potent inhibitors in this study is required to be evaluated in mammalian cells.

3. The authors used NMT1 for the docking, MD simulation, and in vitro enzymatic study, while described the inhibitors as NMT inhibitors. The rationales of NMT1 instead of NMTs should be described, and selectivity of the represent compound is suggested to be investigated.

Minor points:

1. The IC50 values are suggested to be presented in the format of value ± SD in the text and figures.

2. The results of ADME prediction and the removal criteria are suggested to be summarized in one Figure for good readability.

Round 2

Reviewer 2 Report

After revision, the quality of this manuscript was significantly improved, and can reach the required quality standard of Molecules in my opinion. I recommend accepting without further revisions.